# Why the COVID-19 pandemic is a traumatic stressor

**Victoria M. E. Bridgland**[1], **Ella K. Moeck**[1,2], **Deanne M. Green**[1], **Taylor L. Swain**[1], **Diane M. Nayda**[1], **Lucy A. Matson**[1], **Nadine P. Hutchison**[1], **Melanie K. T. Takarangi**[1] *

1 College of Education, Psychology & Social Work, Flinders University, Adelaide, South Australia, Australia,
2 Turner Institute for Brain and Mental Health, Monash University, Melbourne, Victoria, Australia

* melanie.takarangi@flinders.edu.au

**Data Availability Statement:** The authors preregistered this experiment (https://osf.io/dxhek), and together with the data file, all supplementary material—including full demographic breakdown, analyses by country, and

## Abstract

The COVID-19 pandemic does not fit into prevailing Post-traumatic Stress Disorder (PTSD) models, or diagnostic criteria, yet emerging research shows traumatic stress symptoms as a result of this ongoing global stressor. Current pathogenic event models focus on past, and largely direct, trauma exposure to certain kinds of life-threatening events. Yet, traumatic stress reactions to future, indirect trauma exposure, and non-Criterion A events exist, suggesting COVID-19 is also a traumatic stressor which could lead to PTSD symptomology. To examine this idea, we asked a sample of online participants (*N* = 1,040), in five western countries, to indicate the COVID-19 events they had been directly exposed to, events they anticipated would happen in the future, and other forms of indirect exposure such as through media coverage. We then asked participants to complete the Posttraumatic Stress Disorder Checklist-5, adapted to measure pre/peri/post-traumatic reactions in relation to COVID-19. We also measured general emotional reactions (e.g., angry, anxious, helpless), well-being, psychosocial functioning, and depression, anxiety, and stress symptoms. We found participants had PTSD-like symptoms for events that had not happened and when participants had been directly (e.g., contact with virus) *or* indirectly exposed to COVID-19 (e.g., via media). Moreover, 13.2% of our sample were likely PTSD-positive, despite types of COVID-19 "exposure" (e.g., lockdown) not fitting DSM-5 criteria. The emotional impact of "worst" experienced/anticipated events best predicted PTSD-like symptoms. Taken together, our findings support emerging research that COVID-19 can be understood as a traumatic stressor event capable of eliciting PTSD-like responses and exacerbating other related mental health problems (e.g., anxiety, depression, psychosocial functioning, etc.). Our findings add to existing literature supporting a *pathogenic event memory* model of traumatic stress.

## Introduction

The COVID-19 pandemic is ravaging all areas of human life (e.g., social, financial), creating distress, and exacerbating mental health issues [1, 2]. Recent research suggests that *traumatic stress reactions* during the pandemic—including intrusive re-experiencing and heightened arousal—are particularly prevalent [3, 4]. But exposure to the pandemic does not fit neatly within prevailing Post-traumatic Stress Disorder (PTSD) models (pathogenic event models)

other descriptive and inferential statistics—can be found under this project: https://osf.io/jn7zx/.

**Funding:** MKTT and EKM received funding for this research from the College of Education, Psychology and Social Work at Flinders University. The funder had no role in study design, data collection and analysis, decision to publish, or preparation of the manuscript.

**Competing interests:** The authors have declared that no competing interests exist.

[5, 6]. These models, along with the DSM-5 diagnostic criteria [7], attribute traumatic stress reactions to past, and largely direct, exposure to certain life-threatening events, and thus do not readily account for emerging evidence that COVID-19 is associated with PTSD symptomology. We propose that people's traumatic stress reactions to the COVID-19 pandemic may relate more to: the future than the past; indirect (e.g., via media coverage) than direct (e.g., contact with the virus) exposure; and stressful events (e.g., unemployment, isolation, non-sudden illness/death) that do not meet Criterion A (i.e., actual or threatened death, injury or sexual violation). Therefore, we sought evidence for this proposal. Although we know traumatic stress reactions to future, indirect trauma exposure, and non-Criterion A events exist [8, 9], the COVID-19 pandemic gives us a unique opportunity to extend this research by considering all three factors simultaneously. Our goal was not to unnecessarily pathologize normal transient stress reactions [10], but rather to document types of "events" that lead to traumatic stress reactions and thus inform PTSD models which may—currently—not capture all people who require help for traumatic stress symptoms.

We first turn to the idea of traumatic stress as a problem of the past. Existing PTSD models largely focus on traumatic stress as a problem that occurs in response to past, not future, events. Perhaps there is a profound ontological distinction between something that has happened in the past and something that might happen in the future. Yet, Addis ([11], p. 233) argues that remembering and imagining are "fundamentally the same process" (see also [12]); both involve the mental rendering of experience. Certainly, there is convincing evidence that similar neural networks underlie remembering the past and imagining the future [13]. Thus, it is possible that problematic reactions can occur in response to *any* mentally rendered experience, whether it is remembered and/or imagined. Existing evidence fits with this idea: imagining *future* trauma—e.g., among soldiers before deployment, expectant mothers, or people anticipating the death of a loved one—can cause PTSD-related symptoms and distress at similar or higher levels than for past trauma [8, 14–16]. Indeed, participants in these studies have reported—on scales adapted from traditional PTSD measures—experiencing not only more typical symptom candidates like image-based flashforwards to a specific future event, but also nonspecific symptom candidates such as heightened irritability and negative mood. Therefore, traumatic stress symptoms may be a problem of anticipating the future, as well as a reaction to something in the past [8, 15, 16]. Given the unknown timeline of COVID-19, it seems especially likely that PTSD-like symptoms could arise due to anticipating any number of negative future events (e.g., loved ones becoming sick) associated with the virus, particularly in the early weeks of the pandemic (when the current data were collected). Moreover, given COVID-19 is still unfolding, people may experience peri-traumatic reactions (responses at the time of a stressful event or immediately after).

We now examine the idea of traumatic stress arising only from *direct* exposure to a trauma. Early editions of the DSM provided wide and ambiguous definitions of trauma exposure. For example, when PTSD was first introduced in the DSM-III, traumatic events were characterized as catastrophic stressors outside the range of normal human experience [17]. This definition was tightened in the DSM-IV, which stated that people could either witness *or* be "confronted with" events that involved threatened death or serious injury [18]. In an effort to further refine the definition, DSM-5 [7] specifies that a person must directly experience (Criterion A1) or witness a traumatic event (A2). Two exceptions to this rule encompass *indirect* exposure: learning that a loved one has suddenly died in a violent/accidental way (A3), or exposure to others' traumatic experiences (A4: termed "vicarious trauma" e.g., first responders collecting human remains). However, Criterion A4 specifically excludes exposure via electronic media—unless this exposure is *work* related (e.g., police officers repeatedly exposed to child exploitation images). Yet, PTSD symptomology is also found among members of the general public

who are exposed only to traumatic *material*, usually via news media. For example, media exposure was associated with higher acute stress than direct exposure to the Boston Marathon bombings [19]. In China, nurses *not* involved in caring for COVID-19 patients (non-front-line nurses) and the general public had higher PTSD-like symptoms (e.g., intrusive thoughts), depression, anxiety, and stress symptoms, and physiological reactions (e.g., no appetite), than front-line nurses [20]. Li et al. [20] speculated that perhaps non-front-line nurses and the general public consumed more COVID-19 media. Recent evidence supports this interpretation: exposure to COVID-19 related news in the initial stages of the outbreak was associated with negative affect, anxiety, depression and stress [21]. Moreover, searching for additional information about COVID-19 via traditional and social media was related to increased fear about consequences of the virus [22]. Therefore, it seems possible that indirect exposure to the pandemic, such as via the 24-hour news cycle, could produce PTSD-like symptoms.

Last, we turn to the idea of traumatic stress arising only from *exposure to actual or threatened death*, *serious injury or sexual violation* (i.e., only to certain kinds of serious events). Under the current DSM-5 diagnostic criteria, traumatic stress symptoms following events that do not involve an immediate threat to life or physical injury—such as divorce, job loss, or non-sudden medical events—do not qualify for a PTSD diagnosis. Yet, we know PTSD symptoms arise after a range of events that do not meet this narrow definition [9]. For example, ongoing financial stressors and low social support following Hurricane Katrina, not only direct exposure to hurricane-related events, were associated with PTSD symptoms measured 18–24 months post-hurricane [23]. Notably, financial stressors independently predicted PTSD symptom duration, while hurricane-related traumatic events did not. Similarly, receiving a cancer diagnosis does not meet DSM-5 criteria specifying a "sudden and catastrophic" medical event [24]. Yet, 6–20% of cancer sufferers and their families are diagnosed with cancer-related PTSD [24]—although the true prevalence of cancer-related PTSD is likely much higher. Indeed, one of the main reasons that people who met DSM-IV diagnostic criteria do not meet DSM-5 criteria is the elimination of non-sudden death of a loved one as a Criterion A1 event [25]. Thus, direct exposure to a past event that threatens death, injury, or sexual violation is not the only precipitator of PTSD-like symptoms. Again, these data suggest that a range of COVID-19 related stressful events (e.g., job loss, isolation) could lead to traumatic stress symptoms, despite even the direst of COVID-19 events not meeting stringent DSM-5 criteria that state medical events must be sudden and "catastrophic" [7].

In summary, our primary prediction was that people would report experiencing *pre-*, *post-*, and *peri*-traumatic stress reactions in response to COVID-19, regardless of whether they had been directly (e.g., COVID-19 diagnosis) or indirectly (e.g., through media) exposed to the virus, or had other negative experiences (e.g., lockdown). As a secondary aim, we were interested in capturing a snapshot of people's psychological (rather than immunological) response to the virus. We expected that pre/peri/post-traumatic stress reactions, as well as other psychological functioning indicators (well-being, psychosocial impairment, emotions, depression, anxiety, and stress) would fluctuate alongside COVID-19 exposure, and that psychological functioning would vary by demographics (e.g., age, healthcare).

We recruited online participants from five English speaking western countries of similar socio-economic make up (United States, United Kingdom, Canada, Australia and New Zealand). To date, research on COVID-19 as a traumatic stressor has primarily been conducted in China [3], and only one unpublished study has focused on one of our populations of interest (United States [26]). Furthermore, published research to date has focused on providing a snapshot of mental health related concerns (e.g., rates of PTSD, depression etc.), rather than exploring *why* COVID-19 may be considered a traumatic stressor [1, 2]. Here, to investigate why COVID-19 may be a traumatic stressor, we asked participants to indicate the COVID-19

events they had experienced and what future events they were concerned about (see Supplementary Online Material: https://osf.io/jn7zx/), and of these, which event bothered/bothers them the most and why. We also measured COVID-19 media consumption. Participants completed the Posttraumatic Stress Disorder Checklist-5 (PCL-5 [27]), adapted to measure pre/peri/post-traumatic reactions, and measures of general emotional reactions, well-being, psychosocial functioning, and depression, anxiety, and stress symptoms. Although emerging research on COVID-19 and traumatic stress reactions has typically not specified whether participants anchored their reactions to COVID-19 itself (e.g., [28]), we asked our participants to respond to the PCL-5 in relation to COVID-19. In doing so, we reduced the likelihood that participants were reporting general (i.e., unrelated to the pandemic) levels of stress or anxiety.

## Method

This experiment was approved by the Flinders University Social and Behavioural Research Ethics Committee. We have reported all measures, conditions, and data exclusions. We pre-registered this experiment (https://osf.io/dxhek), and together with the data file, all supplementary material—including full demographic breakdown, analyses by country, and other descriptive and inferential statistics—can be found under this project: https://osf.io/jn7zx/.

### Participants

Correlations stabilize when sample size approaches 260 [29]. We excluded 18 responses: five provided answers consistent with bots/farmers (e.g., "interested"), and 13 took the survey twice. Thus, after exclusions, we collected 260 participants in each of four locations (Australia/NZ combined for analysis; a participant error resulted in 261 participants from Canada, and 259 from Australia/NZ), from Mechanical Turk (MTurk; $n = 320$) and Prolific ($n = 720$) between April 10–21, 2020, exceeding the pre-registered sample size ($N = 1,000$). We know that MTurk data are reliable [30, 31] and sometimes even superior to university sourced participants (e.g., participants fail fewer attention checks [32]). Prolific data appear to be as reliable as MTurk, and similar, if not superior to, university research pools [33]. Further, the prevalence of mental health disorders in MTurk populations matches or exceeds that of the general population [34], suggesting MTurk is an excellent source for collecting clinical and subclinical populations. We specifically abide by Mellis and Bickel's [35] advice for conducting online research, to mitigate concerns in these populations (e.g., non-naivete, worker inattention, fraudulent responses and worker treatment). To minimize "bots"/server farmers completing the survey [36, 37], participants had to pass a captcha, a simple arithmetic question (presented as an image to make it difficult for bots to read), and score at least 8/10 on an English proficiency test. We are confident that these entry requirements screened out almost all bots/server farmers; in one estimate, the addition of an English proficiency test screened out 96% of bots/server farmers [38]. In addition to these entry requirements, participants had to pass at least one of three attention check questions embedded in the survey itself to be included in the final sample [39, 40].

Participants ranged from 18–78 years ($M = 35.7$, $SD = 12.3$), half were male (50.70%, female = 48.8%, non-binary = 0.4%; 0.2% did not answer) and most were Caucasian 59.9%. Others were of Asian (13.4%); African (including "Black", 4.7%); Middle Eastern (including "Eurasian", 0.8%); European (3.5%); and Hispanic (1.2%) descent, or Indigenous (0.3%); Pacific Islander (0.1%); Mixed (3.3%) ethnicity. Some participants provided nationality (e.g., "Australian" 12.8%) or no answer (0.2%). Seventy percent were employed, 19.8% were students. Participants' highest level of education was a college/university undergraduate degree (54.8%), postgraduate degree (18.7%), high school (25.6%); 1.0% < high school. Participants had a median of three people in their household (including themselves). Modal household

income was (local currency): Australia/NZ = $100,000-$149,999, Canada = $100,000-$149,000, UK = £20,000-£29,999, US = $50,000-$59,999. Most participants (88.8%) had health-care coverage for COVID-19 expenses and were not in a high-risk group (82.6%) for developing COVID-19 (e.g., elderly, medical condition; 14.9% of participants identified as high risk, 2.5% did not know). Just under half (45.2%) had experienced a Criterion A event.

## Materials

We first asked demographic questions, described above. Note that other pre-registered measures (i.e., media avoidance, engagement with fictional/non-fictional media, perceptions of sharing the pandemic with others) are not reported here.

**General emotional reactions.** We assessed current stress (0 = no stress, 10 = worst stress possible), and, when thinking about COVID-19, how intensely participants felt negative emotions (sad, angry, anxious, frustrated, helpless, fearful, disgusted; 0 = not at all, 7 = extremely). We averaged these emotion items ($\alpha$ = .89).

**Exposure to COVID-19 and related impact.** We provided a list of 32 COVID-19 related events (ranging from exposure to the virus itself to stressors like job loss; see Supplementary Online Material: https://osf.io/tkemz/), within nine predetermined categories. Participants selected all the events they had experienced, with an option for 'other' leading us to create four additional––and modify 10––categories. We recategorized seven "other" responses into new categories and 37 into modified/existing categories. We then re-presented the same list of events, but asked participants to select events they were concerned about happening in the future ("other" events led to three additional categories [seven responses recategorized] and eight modified categories [15 recategorized]). For both lists, participants identified which of their selected events bothered them the most and why (text response), and how much it bothered them emotionally (1 = not at all, 5 = very much).

**Media exposure.** Participants identified the social media (e.g., Facebook) and news media (e.g., television) sources used to engage with COVID-19 content. We assessed frequency of use (how many: hours in the past day, days in the past week, average hours in the past week), and combined these items to create separate scores for both media types.

**The PTSD Checklist (PCL-5 [27]).** Participants rated how much they have been bothered by 20 DSM-5 PTSD symptoms (e.g., "having difficulty concentrating"; 0 = not at all, 4 = extremely; current study $\alpha$ = .92). We made three modifications: measured symptoms in relation to COVID-19 experiences, over the past week (rather than month) due to the rapidly changing circumstances, and asked participants to indicate if each symptom (rated > 0) related to something that happened in the past, was currently happening, or may happen in the future.

**The 5-item World Health Organization Well-Being Index (WHO-5 [41]).** Participants rated how five statements (e.g., "I have felt calm and relaxed") applied to them over the past week (0 = at no time, 5 = all of the time). Total scores (0–25) are multiplied by four to provide a percentage score (0 = worst possible quality of life, 100 = best possible quality of life; current study: $\alpha$ = .90).

**Brief Inventory of Psychosocial Functioning (B-IPF [42]).** Participants rated how much (0 = not at all, 6 = very much) trouble they have had with seven impairment domains (e.g., work) over the past week (e.g., "I had trouble at work"; current study: $\alpha$ = .85).

**Depression, Anxiety and Stress Scale (DASS-21 [43]).** Participants rated the degree to which each statement (e.g., "I felt down-hearted and blue") applied to them over the past week (0 = did not apply to me at all, 3 = applied to me very much). Current study: Depression, $\alpha$ = .92; Anxiety, $\alpha$ = .86; Stress, $\alpha$ = .90.

## Results

For context, over our 12 days of data collection, confirmed cases worldwide increased from ~1.7 to ~2.5 million (deaths from ~102,000 to ~170,000). In the US, total cases jumped from 500,000 to over 750,000, President Trump released the "Opening Up America Again" plan, while protests to remove restrictions increased. In the UK, Prime Minister Johnson was released after hospitalization for COVID-19, the Queen addressed the nation, and lockdown restrictions were extended. In Canada, deaths reached 1,500, and an unrelated shooting occurred in Nova Scotia. In Australia/NZ, lockdown procedures were introduced or maintained, and both countries showed signs of reduced COVID-19 spread from the first wave.

We ran analyses using Null-Hypothesis Significance Tests ($\alpha$ = .05) in SPSS Version 25 and JASP for MacOS version 0.13.1 [44]. Where data were missing, we used the average mean score from the appropriate subscale. In total there were only six missing data points from five participants: two participants missed items on the Posttraumatic Stress Disorder Checklist (PCL-5; one participant missed two items), two participants missed one item on the WHO-5 (measure of wellbeing), and one participant missed one item on the Depression, Anxiety and Stress Scale (DASS-21).

First we present a snapshot of COVID-19's effect on mental health. Descriptive statistics for overall stress, emotional reactions and psychological functioning ratings appear in Table 1.

These variables were highly correlated. According to the conservative cut-off for clinical significance for PCL-5 scores ($<$ 32 = negative, $\geq$ 33 = positive) [45], 13.2% of our participants could be classified PTSD-positive. For *depression*, 47.3% of our participants were in the normal range; 28.8% mild-moderate; 24.0% severe-extremely severe (Depression, Anxiety and Stress Scale [DASS-21] manual cut-offs). For *anxiety*: 68.0% normal, 15.9% mild-moderate, 16.1% severe-extremely severe; and for *stress*: 63.0% normal, 22.6% mild-moderate, 14.4% severe-extremely severe. Participants' mean DASS-21 scores were higher than non-clinical samples [46]. For well-being, 55.4% of participants scored below 50 (0 = worst, 100 = best). Overall well-being was below the UK population mean of 58.6 [47]. For psychosocial functioning, 64.0% of participants reported some impairment due to COVID-19. Completing day-to-day activities (e.g., chores, medical appointments; self-care subscale: 62.8%) was most common. Troubles with training/education (27.6%) created the highest level of impairment ($M$ = 3.3, $SD$ = 1.7).

**Table 1. Descriptive statistics–emotion and psychological functioning variables (full sample).**

|  | Range | Mean | Median | SD |
|---|---|---|---|---|
| Pre/Peri/Post-Traumatic Stress Reactions (PCL-5) | 0–66 | 17.37 | 14.00 | 13.29 |
| Re-experiencing | 0–20 | 3.51 | 2.00 | 3.82 |
| Avoidance | 0–8 | 2.01 | 1.00 | 2.04 |
| Alterations in cog/mood | 0–26 | 6.78 | 6.00 | 5.36 |
| Alterations in arousal | 0–24 | 5.07 | 4.00 | 4.21 |
| Stress | 0–10 | 4.99 | 5.00 | 2.39 |
| Negative emotions | 1–7 | 3.72 | 3.86 | 1.40 |
| Physical Disgust | 1–7 | 3.74 | 4.00 | 1.84 |
| Moral Disgust | 1–7 | 5.33 | 6.00 | 1.57 |
| Wellbeing (WHO-5) | 0–100 | 46.27 | 48.00 | 22.53 |
| Psychosocial Functioning (B-IPF) | 0–100 | 24.42 | 20.00 | 22.43 |
| Depression (DASS-21) | 0–42 | 12.50 | 10.00 | 11.22 |
| Anxiety (DASS-21) | 0–42 | 6.43 | 4.00 | 8.07 |
| Stress (DASS-21) | 0–42 | 12.50 | 10.00 | 10.38 |

Taken together, our findings support emerging research [1, 2, 4] that COVID-19 is associated with PTSD-like symptomology and other mental health issues. Next we addressed our main aim—to examine evidence for our proposal that people are experiencing traumatic stress symptoms as a result of COVID-19 because *direct* exposure to a *past* event that threatens *death*, *injury*, *or sexual violation*, is not the only circumstance capable of precipitating PTSD-like symptoms.

First, we examined evidence that traumatic stress is a problem not only of remembering the past, but also of anticipating the future. We calculated the frequency of exposure to experienced and anticipated COVID-19 events. On average, participants anticipated ($M$ = 8.98, $SD$ = 5.24) more events than they had experienced ($M$ = 6.34, $SD$ = 2.74; $d$ = 0.49). The most common worst experiences were being in lockdown (13.1%), trouble obtaining supplies (11.2%), and voluntarily self-isolating (10.8%). The most common worst anticipated experience was close family/friends passing away (38.1%), followed by becoming ill or passing away (22.2%), and close family/friends testing positive (9.2%). However, worst anticipated and experienced event judgements were fairly evenly distributed across our full list.

We next examined whether traumatic stress reactions were associated with anticipated (and experienced) events. We found small correlations between PTSD-like symptoms and total experienced ($r$ = .20, $p$ < .001) and anticipated ($r$ = .23 $p$ < .001) events. Both exposure measures weakly correlated with other stress and psychological functioning measures (experienced: $r$s = .10-.19; anticipated: $r$s = .12-.22).

However, because examining event totals tells us nothing about how distressing *each* experienced/anticipated event was, we next focused on our nine event categories, and participants' *worst* experienced and anticipated events. Participants who had contact with the virus ($d$ = 0.34, 95% CI [0.20, 0.47]), lost work/income ($d$ = 0.18 [0.05, 0.30]), experienced lockdown directives ($d$ = 0.13 [0.004, 0.25]), changes to children/dependents ($d$ = 0.15 [0.02, 0.28]), and had trouble buying supplies ($d$ = 0.34 [0.22, 0.46]) reported higher PTSD symptoms than participants not reporting these experiences. However, there was no difference based on whether people experienced changes to work/education, travel plans, closure of non-essential gatherings, or voluntarily self-isolated ($d$s 0.02–0.11). Similarly, participants who anticipated contact with the virus ($d$ = 0.44, 95% CI [0.15, 0.72]), losing work/income ($d$ = 0.25 [0.12, 0.37]), lockdown directives ($d$ = 0.15 [0.03, 0.28]) and changes to children/dependents ($d$ = 0.35 [0.18, 0.51]) had higher PTSD symptoms than participants who did not anticipate these events. Contrary to experienced events, participants who anticipated changes to work/education ($d$ = 0.38, [0.16, 0.61]), travel plans ($d$ = 0.26 [0.12, 0.40]), and closure of non-essential gatherings ($d$ = 0.29 [0.07, 0.50]) reported higher PTSD symptoms than participants who did not anticipate these events. There was no difference based on whether people anticipated voluntarily self-isolating ($d$ = 0.24 [0.01, 0.46]), or having trouble buying supplies ($d$ = 0.12 [-0.02, 0.25]).

On average, participants' worst anticipated event ($M$ = 3.97, $SD$ = 1.04) was more emotionally bothersome than their worst experienced event ($M$ = 3.35, $SD$ = 1.15; two-tailed paired samples: $t$ (1039) = -18.18, $p$ < .001, $d$ = 0.56, 95% CI [0.50, 0.63]). Both ratings strongly correlated with PTSD symptoms (experienced: $r$ = .48; anticipated: $r$ = .45) and other stress and psychological functioning measures (experienced: $r$s .30-.51; anticipated: $r$s .25 - .46). However, total PCL score does not tell us how often participants *attributed* symptoms to experienced and anticipated events, or something currently happening. Nearly half (48.1%) of participants reported symptoms in relation to all time periods (23.3% past/future, 7.5% past/current, 0.8% past/future; 14.5% current, 0.2% past, 1.7% future; 3.9% reported no symptoms). Some participants failed to index individual symptoms to a timeframe; 0.6% did not index any. Notably, 73.5% of participants reported at least one symptom related to something that had not happened. These symptoms fell into all four symptom clusters (50.1% of participants attributed

intrusion symptoms to the future; 15.4% avoidance, 48.0% hyperarousal and 56.5% cognition and mood). In other words, PTSD-like symptoms for events that had not yet happened did not only manifest as intrusive thoughts (which might be narrowly conceptualized as worries about the future) but also as nonspecific symptoms such as irritability and alertness.

Second, we examined whether traumatic stress reactions occurred regardless of direct versus indirect exposure to COVID-19. Participants reporting PTSD symptoms in relation to *future* events already supports this prediction. In addition, we found small correlations between COVID-19 related social media consumption ($r = .18$, $p < .001$) and PTSD symptoms, and all psychological functioning variables ($r$s = .09 - .20) except well-being; and between traditional media consumption and PTSD symptoms ($r = .12$, $p < .001$), stress, and negative emotions ($r$s = .14, $p < .001$), but not other psychological functioning variables. Although small, considering the average adult spends three hours and 30 minutes a day on a mobile device—equating to 53 days in a year and countless exposure to news media [48]— these effects could rapidly accumulate, leading to adverse consequences [49].

Third, we examined evidence that PTSD-like symptoms occur for events that do not involve actual or threatened death, injury, or sexual violation. Arguably, *none* of our events/ categories meet Criterion A; medically-based trauma is limited to sudden catastrophe (e.g., waking during surgery, anaphylactic shock [7]). Even our most extreme direct exposure variables (e.g., being hospitalized in a critical condition) do not qualify. Therefore, it is troubling that 13.2% of participants could be classified as PTSD-positive when anchoring the PCL-5 to COVID-19. We explored whether PTSD-positive participants had direct exposure to the virus via events which, while not classed as Criterion A, could be life threatening. Direct exposure included personally testing positive or being hospitalized as a result of COVID-19, or suspecting you may have been exposed to the virus, or knowing close others (e.g., family/friends) who have tested positive, were hospitalized, or who have passed away, as a result of COVID-19. Overall, of the 13.2% of participants who were classed as PTSD-positive, 60.6% were participants who were *not* directly exposed to the virus (versus those directly exposed; 39.4%; $\chi^2$ (1) = .031, $\varphi = .067$). However, it should be noted that roughly double the number of participants had been indirectly, rather than directly exposed to the virus. Indeed, when examining within these groups, more participants who had some form of direct virus exposure ($n = 327$) were PTSD-positive (16.5%), than participants who had no virus exposure ($n = 713$; PTSD-positive = 11.6%). These data suggest that having direct contact with the virus (e.g., one's self or a family member being diagnosed) did result in increased PTSD-like symptomology versus people who were only indirectly exposed (e.g., affected by lockdown). Additionally, there was no difference in the percentage of PTSD-positive participants who had experienced (6.8%) or not (6.3%) a Criterion A event previously, suggesting participants anchored PCL-5 responses to COVID-19 experiences rather than other lifetime traumas.

Together, these data support our proposal that PTSD-like reactions can occur in relation to past, ongoing, and future events, indirect event exposure, and for a variety of stressors not covered by Criterion A. To examine the relative contribution of our exposure variables, while also controlling for demographic variables that could increase participants' susceptibility to PTSD symptoms, we regressed PCL-5 total on these variables. For all regression analyses, tolerance and variance inflation factors showed multicollinearity was not an issue.

First, we examined which demographic variables related to PTSD-like reactions. PTSD symptomology (PCL-5 total scores) was higher for participants from the United Kingdom (vs. Australia/New Zealand: $d = 0.35$, 95% CI [-0.52, -0.17], and vs. the United States: $d = 0.45$, [0.27, 0.62]), and Canada (vs. United States: $d = 0.33$, [0.16, 0.51]). Further, scores were higher for females ($d = 0.41$, [0.29, 0.54]), students ($d = 0.16$, [0.01, 0.32]), people at high risk of contracting COVID-19 ($d = 0.30$, [0.14, 0.46]), and people with lifetime Criterion A trauma

($d$ = 0.31, [0.18, 0.43]), and were weakly correlated with age and income ($rs$ < .10). For categorical variables, dummy variables were created as per the protocol Field [50] describes. For country of residence, we selected the United States as the baseline group (i.e., the group which is always assigned 0 in each dummy variable) because participants from the United States reported the lowest level of PTSD symptoms. After entering these variables into a single step of a simultaneous regression, we found that country of residence, age, sex, risk status, and prior Criterion A exposure best predicted PTSD symptomology (10.2% of the variance; $F$(9, 1031) = 13.95, $p$ < .001).

To examine the unique additional variance explained by actual and anticipated exposure to COVID-19 (event totals), media consumption (social and traditional media summed), and the emotion associated with worst events (experienced and anticipated), we then ran a hierarchical regression with demographic predictors (identified in our simultaneous regression) entered at Step 1, exposure variables at Step 2, and emotion variables at Step 3. After controlling for demographics, exposure variables explained an additional 7.1% of the variance in PTSD symptomology ($F_{change}$ (3, 1020) = 29.23, $p$ < .001). Importantly, emotion associated with worst events explained an additional 16.2% of the variance in PTSD symptomology ($F_{change}$ (2, 1018) = 125.26, $p$ < .001; regression coefficients and standard errors appear in Table 2). The final model explained 33.3% of the variance in PTSD symptomology ($F$ (12, 1030) = 43.86, $p$ < .001), and, notably, experienced event totals was no longer a significant predictor. In sum, the emotion associated with participants' worst experienced and worst anticipated events was the strongest predictor of PTSD symptomology, above and beyond demographic variables and measures of direct and indirect exposure to COVID-19 related events.

## Discussion

Overall, we found that participants had PTSD-like symptoms for events that had not yet happened, challenging the nature of traumatic stress as a problem pertaining only to the past. Participants reported these reactions whether they had been directly (e.g., COVID-19 diagnosis) *or* indirectly exposed (e.g., via media) to COVID-19, challenging the idea that people need to experience a direct, in person event to develop PTSD-like symptoms. Finally, 13.2% of our sample were PTSD-positive, despite COVID-19 "exposure" not fitting within Criterion A. The emotion associated with participants' *worst* experienced and anticipated events predicted PTSD-like symptoms *beyond* demographic and exposure variables, suggesting that subjective evaluations of emotional impact may be more important in determining traumatic stress reactions than individual characteristics or objective levels of exposure.

Dominant pathogenic event models focus on PTSD arising for specific kinds of external events, and emphasize factors during/after encoding, including differential processing for perceptual (e.g., sensory) versus conceptual (e.g., meaning) details, and poor integration of the trauma in autobiographical memory [5, 6]. Thus, they do not account for symptoms in response to anticipated events, indirect exposure to trauma, or events that do not meet Criterion A. Our findings fit instead with a *pathogenic event memory* model, which accounts for traumatic stress in response to future and/or imagined events [51].

Of course, an alternative explanation of our data for anticipated events is that they simply reflect people expressing distress—including negative or threatening thoughts and images— about potential threats looming in their future, which may reflect worry, or a generalized anxiety disorder [7, 52]. Indeed, the PCL-Civilian scale—which is indexed to "stressful life experiences" rather than a specific trauma—correlates highly with measures of depression and general anxiety, suggesting that without the anchor to a specific trauma, the PCL-5 may pick up negative emotionality more generally [53]. However, we did ask people to indicate specific

**Table 2. Regression analysis summary for demographic factors, exposure variables, and the emotion associated with worst events (experienced and anticipated) predicting PCL total scores.**

| Model | | *B* | *95% CI* | *B* |
|---|---|---|---|---|
| Step 1 | Constant | 25.49 | [20.97, 30.01] | |
| | Age | -0.16 | [-0.23, -0.08] | -.15** |
| | Household income | -0.07 | [-0.32, -0.18] | -.02 |
| | Sex | 4.01 | [2.42, 5.60] | .15** |
| | Risk status | -4.71 | [-6.87, -2.55] | -.13** |
| | Lifetime trauma exposure | -3.30 | [-4.88, -1.71] | -.12** |
| | Student status | -0.33 | [-2.53, 1.87] | -.01 |
| | Australia/NZ vs. US | -0.18 | [-2.52, 2.15] | -.01 |
| | Canada vs. US | 2.60 | [0.31, 4.88] | .08* |
| | UK vs. US | 5.34 | [3.08, 7.59] | .17** |
| Step 1 | Constant | 25.20 | [20.95, 29.5] | |
| | Age | -0.16 | [-0.23, -0.09] | -.15** |
| | Sex | 3.98 | [2.40, 5.56] | .15** |
| | Risk status | -4.75 | [-6.89, -2.61] | -.14** |
| | Lifetime trauma exposure | -3.37 | [-4.96, -1.79] | -.13** |
| | Australia/NZ vs. US | -0.21 | [-2.48, 2.07] | -.01 |
| | Canada vs. US | 2.49 | [0.24, 4.74] | .08* |
| | UK vs. US | 5.37 | [3.17, 7.58] | .18** |
| Step 2 | Constant | 10.79 | [5.51, 16.07] | |
| | Age | -0.16 | [-0.23, -0.09] | -.15** |
| | Sex | 3.62 | [2.09, 5.16] | .14** |
| | Risk status | -4.22 | [-6.29, -2.16] | -.12** |
| | Lifetime trauma exposure | -2.62 | [-4.17, -1.07] | -.10* |
| | Australia/NZ vs. US | -0.25 | [-2.45, 1.95] | -.01 |
| | Canada vs. US | 2.05 | [-0.12, 4.22] | .07 |
| | UK vs. US | 4.63 | [2.49, 6.76] | .15** |
| | Media consumption total | 0.32 | [0.20, 0.43] | .16** |
| | Total experienced events | 0.38 | [0.09, 0.67] | .08* |
| | Total anticipated events | 0.45 | [0.30, 0.59] | .18** |
| Step 3 | Constant | -6.71 | [-12.01, -1.41] | |
| | Age | -0.10 | [-0.16, -0.43] | -.10* |
| | Sex | 1.36 | [-0.05, 2.76] | .05 |
| | Risk status | -2.80 | [-4.66, -0.94] | -.08* |
| | Lifetime trauma exposure | -2.60 | [-3.99, -1.21] | -.10** |
| | Australia/NZ vs. US | 0.37 | [-1.60, 2.35] | .01 |
| | Canada vs. US | 1.88 | [-0.07, 3.84] | .06 |
| | UK vs. US | 4.26 | [2.35, 6.18] | .14** |
| | Media consumption total | 0.16 | [0.06, 0.27] | .08* |
| | Total experienced events | -0.01 | [-0.27, 0.26] | <-.01 |
| | Total anticipated events | 0.26 | [0.12, 0.39] | .10** |
| | Experienced worst event emotion | 3.30 | [2.61, 3.99] | .28** |
| | Anticipated worst event emotion | 3.04 | [2.28, 3.81] | .23** |

Note

* $p < .005$

** $p < .001$

events that had happened to them, or that they were concerned about happening in the future, and then index their symptoms in relation to COVID-19. Therefore, it seems likely participants were thinking about specific COVID-19 related events rather than unknown/non-specific future events. Asking participants to index their symptoms to COVID-19 related events is key, since symptoms of PTSD are focused on specific events, whereas Generalized Anxiety Disorder involves feelings of anxiety across a wide range of circumstances and situations (e.g., about health or financial concerns). Indeed, hallmark symptoms of PTSD such as intrusions or avoidance have intentionality or "aboutness" [54]. That is, the intrusive thoughts are about something, and the person is avoiding reminders of something—that something being the traumatic event [54].

Our findings may also reflect the "hedonic treadmill" [55], whereby mildly stressful events feel more negative against the backdrop of relatively stress-free lives, particularly in the context of data from western countries, as we measured here. This explanation does not diminish that our participants were experiencing genuine distress, but explains why exposure to events like government lockdowns and inability to source supplies was associated with traumatic stress symptoms—and in some cases, above clinical cut-off levels.

Although our data suggest that COVID-19 could be understood as a traumatic stressor, this conceptualization might contribute to the problem of "conceptual bracket creep" [10]. Thus, we must consider the difference between genuine symptoms of a disorder and normal stress reactions. Failure to do so may undermine efforts to understand the psychological mechanisms that contribute to PTSD symptoms. Indeed, while here we aimed to advance understanding of why COVID-19 may be a traumatic stressor and how it may be understood using existing PTSD frameworks, other researchers have taken a step further, advocating for COVID-19 unique disorders such as "COVID stress syndrome" [56] and "COVID-19 phobia" [57]. Given the novel nature of the COVID-19 pandemic, and its continuing global impact, there is little doubt that the way we characterize the psychological impact of COVID-19 will change as our understanding grows.

COVID-19's psychological fallout has been dubbed the "second curve," predicted to last for months to years. Notably, while most of our participants reported experiencing some form of psychological distress and 13.2% of our sample were likely PTSD positive when anchoring symptoms to COVID-19, only 2% of our total sample reported they had personally tested positive to COVID-19, and only 5% reported that close family and friends had tested positive. It therefore seems likely that the psychological fallout from COVID-19 may reach further than the medical fallout. Short term, our findings highlight the need to focus on the acute psychological distress—especially the perceived emotional impact of particular events—associated with COVID-19 and support other emerging research that demonstrates the damaging psychological impact of COVID-19 on mental health [1–4]. Our results also support the WHO recommendation [58] to minimize consumption of COVID-19 media. Long-term, comprehensive documentation of COVID-19 related traumatic stress reactions will allow health professionals to help people who could otherwise fall through the cracks.

Our research has limitations. First, we used the PCL-5, a self-report measure, to examine PTSD-like symptoms cross-sectionally. The Clinician Administered PTSD Scale [59] is a more comprehensive measure that identifies the frequency and intensity of PTSD-like symptoms for an indexed event and allows a clinician to diagnose if PTSD might be present. However, we wanted to capture PTSD-like symptoms in a large sample across multiple locations that were at different points of the pandemic at the time, a goal that could not be achieved using a clinician administered scale. Second, the Clinician Administered PTSD Scale can only be completed on clients who have experienced a Criterion A event. Here, our aim was to examine traumatic stress reactions to any type of COVID-19 event that might produce such symptoms,

rather than events specifically falling within this category. Similarly, we cannot determine whether symptom levels in our sample would qualify for a PTSD diagnosis. Future research could use experience-sampling methods to capture longitudinal symptoms, or a clinician administered scale to diagnose PTSD.

Second, our sample only examined five Western, English speaking countries similar in culture and socio-economic status. As such, we can only cautiously generalize these findings to other parts of the world. Third, although our data provide evidence for the potential psychological fallout of the COVID-19 pandemic, we did not investigate any personality traits, or external factors such as social support or other coping strategies, that might moderate this response. Finally, we only asked participants to focus on negative outcomes of COVID-19; future research could examine positive outcomes (e.g., growth, resilience).

COVID-19 related traumatic stress symptoms expose weaknesses in dominant PTSD models, which fail to account for traumatic stress reactions arising from anticipated, indirect, or non-life threatening stressors. This study clearly demonstrates that a single global event can lead to PTSD symptoms for a range of anticipated, indirect, and non-life threatening stressors. As Horesh and Brown [60] state, there is a clear need to expand PTSD models and in turn, better capture *all* people who need help for traumatic stress symptoms.

## Author Contributions

**Conceptualization:** Victoria M. E. Bridgland, Ella K. Moeck, Deanne M. Green, Taylor L. Swain, Diane M. Nayda, Lucy A. Matson, Nadine P. Hutchison, Melanie K. T. Takarangi.

**Data curation:** Victoria M. E. Bridgland, Ella K. Moeck, Deanne M. Green, Taylor L. Swain, Diane M. Nayda, Lucy A. Matson, Nadine P. Hutchison, Melanie K. T. Takarangi.

**Formal analysis:** Victoria M. E. Bridgland, Ella K. Moeck, Deanne M. Green, Taylor L. Swain, Diane M. Nayda, Lucy A. Matson, Nadine P. Hutchison, Melanie K. T. Takarangi.

**Methodology:** Victoria M. E. Bridgland, Ella K. Moeck, Deanne M. Green, Taylor L. Swain, Diane M. Nayda, Lucy A. Matson, Nadine P. Hutchison, Melanie K. T. Takarangi.

**Writing – original draft:** Victoria M. E. Bridgland, Ella K. Moeck, Melanie K. T. Takarangi.

**Writing – review & editing:** Victoria M. E. Bridgland, Ella K. Moeck, Deanne M. Green, Taylor L. Swain, Diane M. Nayda, Lucy A. Matson, Nadine P. Hutchison, Melanie K. T. Takarangi.

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
