## [Decision Letter · Decision Letter 0]

13 Nov 2020

PONE-D-20-29313

Why the COVID-19 pandemic is a traumatic stressor

PLOS ONE

Dear Dr. Takarangi,

Thank you for submitting your manuscript to PLOS ONE. After careful consideration, we feel that it has merit but does not fully meet PLOS ONE’s publication criteria as it currently stands. Therefore, we invite you to submit a revised version of the manuscript that addresses the points raised during the review process.

We look forward to receiving your revised manuscript.

Kind regards,

Vedat Sar, M.D.

Academic Editor

PLOS ONE

Journal Requirements:

2) We note that you have stated that you will provide repository information for your data at acceptance. Should your manuscript be accepted for publication, we will hold it until you provide the relevant accession numbers or DOIs necessary to access your data. If you wish to make changes to your Data Availability statement, please describe these changes in your cover letter and we will update your Data Availability statement to reflect the information you provide.

Reviewers' comments:

Reviewer's Responses to Questions

**Comments to the Author**

1. Is the manuscript technically sound, and do the data support the conclusions?

Reviewer #1: Yes

Reviewer #2: Partly

2. Has the statistical analysis been performed appropriately and rigorously? 

Reviewer #1: Yes

Reviewer #2: Yes

3. Have the authors made all data underlying the findings in their manuscript fully available?

Reviewer #1: Yes

Reviewer #2: Yes

4. Is the manuscript presented in an intelligible fashion and written in standard English?

Reviewer #1: Yes

Reviewer #2: No

5. Review Comments to the Author

Reviewer #1: This is a well-organized manuscript which tries to build an alternative model of traumatic stress. They argue that traumatic stress reactions to the Covid-19 may relate more to future than past, indirect than direct exposures and stressful events that do not meet Criterion A.

There are a number of issues with the analysis and the interpretation of results that need to be addressed.

1. Within the flow of the introduction, the 3 features mentioned above have been discussed in order, but the cancer sample in P.5 117-1222 seems to be out of line.

2. Although it is understood from the whole of the introduction part that the authors aimed to determine the frequency of covid-related pre/post/peri traumatic stress reactions in the relevant population and to identify the factors that determine the severity of this stress, I recommend adding the primary and secondary objectives or hypothesis of the study clearly and explicitly.

3. I understand that 45.2% of the population had a Criterion A event in the past which reminds me that the index population can not be defined as a non-clinic population. It would be better to mention this unforeseeable feature as a limitation in the discussion.

4. The time interval should be specified in terms of having a sense of environmental and social conditions in that countries.

5. I wonder if the mean PCL-5 scores were significantly different between those 5 countries.

In this study, collecting data from 5 countries at different epidemic stages is an advantage in terms of generalizing the results, however, if the PCL-5 scores differ between countries, the geographical location should be included in the regression model as a variable.

6. A detailed statistical analysis part is missing within the Methods section. Especially the regression method used for predictive modeling should be added.

7. The prevalence of PTSD is 16.5% in direct exposure group whereas this rate is 11.5% in indirect exposure group.

The reason for being the majority of those with PTSD positive had indirect exposure is because the rate of indirect exposure in the sample is too high (direct exposure 327, indirect exposure 713). So, it is hard to say subjects with indirect exposure had similar PTSD rates with subjects who had direct exposures.

8. As you mentioned in P.19 414 the anticipated events may reflect generalized anxiety disorder. Do you have any explanatory hypothesis as to how these two concepts differ from each other?

Best Regards.

Reviewer #2: This study discusses COVID-19 as a traumatic stressor and focus on DSM-5 and its definitions. The study includes participants from different countries. The sample includes a good age range and authors describe how they accessed the participants and how they calculated the sample size. However, the manuscript is difficult to read and not well organized. In addition, there are many recent studies that study COVID-19 process as traumatic stress and this manuscript does not mention about them and does not discuss their findings and methodology compared to that studies.

As other important points,

1. The abstract is difficulty to read and does not present much information about the methodology and results. I suggest that the authors re-write the abstract and describe their methodology and results in more detail.

2. Introduction includes repeated explanations and needs to be tightened up.

3. Results and discussion sections are combined, which makes it difficult to follow. Statistical analysis are also reported in this section. They should be seperated with more specific subheadings.

4. In line 275, it is not clear how authors reach to such a definite conclusion with the previous reported result.

5. Authors do not discuss the current literature on the topic, but just discuss their findings from their own point of view.

6. Authors do not discuss the traumatic stress definitions done before the introduction of DSM-5.

6. PLOS authors have the option to publish the peer review history of their article (what does this mean?). If published, this will include your full peer review and any attached files.

Reviewer #1: No

Reviewer #2: No

---

## [Author Response · Author response to Decision Letter 0]

26 Nov 2020

Please see the "Response to Reviewers" document, per instructions in the decision letter.

---

## [Decision Letter · Decision Letter 1]

28 Dec 2020

Why the COVID-19 pandemic is a traumatic stressor

PONE-D-20-29313R1

Dear Dr. Takarangi%,

We’re pleased to inform you that your manuscript has been judged scientifically suitable for publication and will be formally accepted for publication once it meets all outstanding technical requirements.

The only point I would recommend in the final version of the manuscript is, as the reviewer advices, to present the statistical analysis under a separate heading instead of being detailed within the conclusion section for ease of reading.

Kind regards,

Vedat Sar, M.D.

Academic Editor

PLOS ONE

Additional Editor Comments (optional):

As the reviewer recommends, n the final version of the manuscript, please present the statistical analysis under a separate heading instead of being detailed within the conclusion section for ease of reading.

Reviewers' comments:

Reviewer's Responses to Questions

**Comments to the Author**

1. If the authors have adequately addressed your comments raised in a previous round of review and you feel that this manuscript is now acceptable for publication, you may indicate that here to bypass the “Comments to the Author” section, enter your conflict of interest statement in the “Confidential to Editor” section, and submit your "Accept" recommendation.

Reviewer #1: All comments have been addressed

2. Is the manuscript technically sound, and do the data support the conclusions?

Reviewer #1: Yes

3. Has the statistical analysis been performed appropriately and rigorously? 

Reviewer #1: Yes

4. Have the authors made all data underlying the findings in their manuscript fully available?

Reviewer #1: Yes

5. Is the manuscript presented in an intelligible fashion and written in standard English?

Reviewer #1: Yes

6. Review Comments to the Author

Reviewer #1: All raised points have been addressed properly by authors. The only point I would recommend in the final version of the manuscript is to present the statistical analysis under a separate heading instead of being detailed within the conclusion section for ease of reading.

7. PLOS authors have the option to publish the peer review history of their article (what does this mean?). If published, this will include your full peer review and any attached files.

Reviewer #1: No

---

## [Editor Report · Acceptance letter]

2 Jan 2021

PONE-D-20-29313R1 

Why the COVID-19 pandemic is a traumatic stressor 

Dear Dr. Takarangi:

I'm pleased to inform you that your manuscript has been deemed suitable for publication in PLOS ONE. Congratulations! Your manuscript is now with our production department. 

Kind regards, 

on behalf of

Dr. Vedat Sar 

Academic Editor

PLOS ONE